# PHOSPHATE STARVATION RESPONSE transcription factors enable arbuscular mycorrhiza symbiosis

Debatosh Das [1,2], Michael Paries [3], Karen Hobecker [3], Michael Gigl [4], Corinna Dawid [4], Hon-Ming Lam [2,5], Jianhua Zhang [2,5,6✉], Moxian Chen [1✉] & Caroline Gutjahr [3✉]

Arbuscular mycorrhiza (AM) is a widespread symbiosis between roots of the majority of land plants and *Glomeromycotina* fungi. AM is important for ecosystem health and functioning as the fungi critically support plant performance by providing essential mineral nutrients, particularly the poorly accessible phosphate, in exchange for organic carbon. AM fungi colonize the inside of roots and this is promoted at low but inhibited at high plant phosphate status, while the mechanistic basis for this phosphate-dependence remained obscure. Here we demonstrate that a major transcriptional regulator of phosphate starvation responses in rice PHOSPHATE STARVATION RESPONSE 2 (PHR2) regulates AM. Root colonization of *phr2* mutants is drastically reduced, and PHR2 is required for root colonization, mycorrhizal phosphate uptake, and yield increase in field soil. PHR2 promotes AM by targeting genes required for pre-contact signaling, root colonization, and AM function. Thus, this important symbiosis is directly wired to the PHR2-controlled plant phosphate starvation response.

[1] State Key Laboratory Breeding Base of Green Pesticide and Agricultural Bioengineering, Key Laboratory of Green Pesticide and Agricultural Bioengineering, Ministry of Education, Research and Development Center for Fine Chemicals, Guizhou University, Guiyang, China. [2] CUHK Shenzhen Research Institute, No. 10 Yuexing 2nd Road, Nanshan, Shenzhen, China. [3] Plant Genetics, TUM School of Life Sciences, Technical University of Munich (TUM), Emil Ramann Str. 4, 85354 Freising, Germany. [4] Chair of Food Chemistry and Molecular Sensory Science, TUM School of Life Sciences, Technical University of Munich (TUM), Lise-Meitner-Str. 34, D-85354 Freising, Germany. [5] State Key Laboratory of Agrobiotechnology, The Chinese University of Hong Kong, Shatin, Hong Kong. [6] Department of Biology, Hong Kong Baptist University, Shatin, Hong Kong. ✉email: jzhang@hkbu.edu.hk; cmx2009920734@gmail.com; caroline.gutjahr@tum.de

Phosphate ($P_i$) is vital for plant growth, yield, and survival, but in most soils, it is poorly available to plants. To facilitate $P_i$ acquisition, 80% of land plants associate with *Glomeromycotina* fungi in a 450-million-year-old arbuscular mycorrhiza (AM) symbiosis, in which the fungus provides mineral nutrients in exchange for carbon[1]. AM symbioses also improve plant stress resistance and soil stability, making AM a promising addition to sustainable agricultural practices[2]. Symbiotic plants acquire most of their phosphate through AM fungi (AMF)[3]. The fungi collect $P_i$ via a subterranean hyphal network and release it inside the root cortex via tree-shaped arbuscules[1]. It has been observed for decades that AM symbiosis is suppressed when the plant $P_i$ status is high, likely to conserve carbon[4–6]. However, the molecular mechanisms interconnecting plant $P_i$ status with AM symbiosis development and function have largely remained elusive. Under $P_i$ deficiency, plants activate a range of adaptive responses including transcriptomic, metabolic, and developmental changes[7]. A majority of these is controlled by PHOSPHATE STARVATION RESPONSE (PHR) proteins, first discovered in the AM-incompetent species *Arabidopsis thaliana* and the unicellular alga *Chlamydomonas reinhardtii*[8]. PHRs belong to the MYB transcription factor (TF) family and bind to PHR1 binding site (P1BS) motifs (GNATATNC) in the promoters of phosphate starvation-induced (PSI) genes[8,9]. Interestingly, in *Lotus japonicus*, P1BS motifs have also been found in the promoters of several genes induced during arbuscule development in root cortex cells[10,11], often in combination with CTTC/MYCS motifs, which are bound by an AM-induced WRINKLED (WRI,) TF[11,12]. Therefore, we hypothesized that PHR may regulate $P_i$ responsive AM development[13]. In contrast to *Arabidopsis*, *L. japonicus* and the important staple crop rice form AM symbioses[14]. The rice genome contains four genes encoding PHR proteins (*PHR1-4*), but based on promoter:*GUS* activity, *PHR2* is most broadly expressed in roots and especially in tissues relevant for AM[15,16]. Thus, we focused on PHR2 in rice to investigate the role of this central phosphate starvation response regulator in AM.

We demonstrate that rice PHR2 is required for root colonization by AMF. Root colonization of *phr2* mutants is drastically reduced at low $P_i$, while ectopic *PHR2* expression partially rescues root colonization at high $P_i$. Even in non-colonized roots, the expression of almost 70% of genes that have been genetically shown to be relevant for AM development and function or that are induced during AM, is reduced in *phr2* mutants compared to wild type. PHR2 targets essential genes for major and indispensable symbiotic processes, i.e. pre-contact signal exchange between the symbionts, accommodation of the fungus inside the root, and symbiotic nutrient exchange. In addition, *PHR2* is required for root colonization, mycorrhizal phosphate uptake, and AM-mediated yield-promotion in field soil and is sufficient to promote root colonization at high $P_i$. Moreover, the role of PHRs in promoting AM is conserved in the legume *L. japonicus*. Thus, PHRs support AM development and function as part of the plant phosphate starvation response syndrome. At high phosphate AM cannot develop due to repression of PHR activity through so-called SPX proteins[17,18]. AM symbiosis is thus directly wired to the plant phosphate starvation response via the master regulator PHR.

## Results

**PHR2 promotes root colonization by AMF.** We examined root colonization by the model AMF *Rhizophagus irregularis* of wild type and *phr2* mutants at low phosphate conditions (25 μM $P_i$; LP). Wild type roots were well colonized, but root colonization was severely impaired in a *phr2* insertion mutant[17] and an independent CRISPR-Cas9 generated *phr2* mutant ('*phr2*(C)[19];

Fig. 1A–D). We observed very few colonization attempts on mutant roots, mainly comprising hyphopodia or very few intraradical hyphae, resembling mutants in common symbiosis genes that encode proteins of a signaling cascade which is critically required for fungal root entry and AM development[14]. Only in very rare colonization units, arbuscules with wild-type-like morphology developed in the *phr2* mutants (Fig. 1E–H and Supplementary Fig. 1).

As PHR2 is critical for root colonization at LP, we examined whether ectopic expression of *PHR2* driven by the 35S promoter (*35S:PHR2*)[17] enables root colonization at higher $P_i$ conditions. At LP, total colonization, arbuscules, and vesicles were slightly increased in the *35S:PHR2* line as compared to wild type. However, at medium (200 μM $P_i$, MP) and high $P_i$ (500 μM $P_i$, HP) wild type and *phr2* were hardly colonized, while *35S:PHR2* enabled 40 and 20% total root length colonization and 25 and 5% colonization with arbuscules, respectively (Fig. 1A–D and Supplementary Figs. 2 and 3). Together this indicates that PHR2 promotes AM development at $P_i$ starvation and its inactivity may be responsible for the absence of AM at high phosphate, as described for other phosphate starvation responses[17,20]. The role of PHR2 in supporting arbuscule development is supported by the activity pattern of its promoter. Shi et al. showed that the *PHR2* promoter is active across all tissue layers in non-colonized rice roots. In colonized roots, a stronger promoter activity was observed in arbuscule-containing cells with an overall increased GUS activity in large lateral roots[18].

**PHR2 activates AM-relevant genes.** To explain the role of PHR2 in AM, we performed RNA sequencing for non-inoculated (Mock) and *R. irregularis*-inoculated (AM) roots of *phr2* at LP, *35S:PHR2* at HP, and wild type at both $P_i$ conditions (Fig. 2 and Supplementary Fig. 4, and Supplementary Data 1). A larger number of genes differed in expression in *phr2* or *35S:PHR2* vs. wild type in AM compared to Mock roots (Fig. 2A). The drastic transcriptome modulation upon AM indicated by principal component analysis in the wild type at LP was clearly reduced for *phr2* (Fig. 2B). At HP, the transcriptomes of all *35S:PHR2* and wild-type samples clustered together, but they separated when analyzed separately from LP samples (Supplementary Fig. 5), suggesting a weaker but still remarkable effect of ectopic *PHR2* expression on the AM root transcriptome at HP. Interestingly, the expression of AM-relevant genes is already compromised in *phr2* Mock roots (without root colonization), similar to wild type at HP (Fig. 2C–F and Supplementary Data 2), including 16 genes with genetically determined and (for most of them) essential functions in AM (Fig. 2E). These include genes encoding receptors involved in the perception of fungal signals prior to root contact (*CERK1*, *NFR5*), common symbiosis genes involved in signal transduction and regulation of colonization (*SYMRK*, *CCaMK*, *CYCLOPS*, *SLR1/DELLA*, *NSP1*, *NSP2*, and *VAPYRIN*), genes involved in strigolactone biosynthesis for activation of the fungus in the rhizosphere prior to contact (*CCD7*, *CCD8B*), genes involved in the regulation of colonization (*KIN2*), genes responsive to karrikin signaling and involved in apocarotenoid biosynthesis (*DLK2C*, *ZAS*), and genes encoding periarbuscular membrane-localized ABCG and phosphate transporters for nutrient exchange (*STR1*, *PT11*, and *PT13*)[21,22]. In addition, the expression of one-fifth of AM-induced genes in wild type is reduced already in the absence of AMF in *phr2* (Fig. 2F and Supplementary Data 3). Together, this suggests that PHR2 conditions the root for AM symbiosis at low phosphate and that the lower cumulative expression of AM-relevant genes impairs AM in *phr2* mutants.

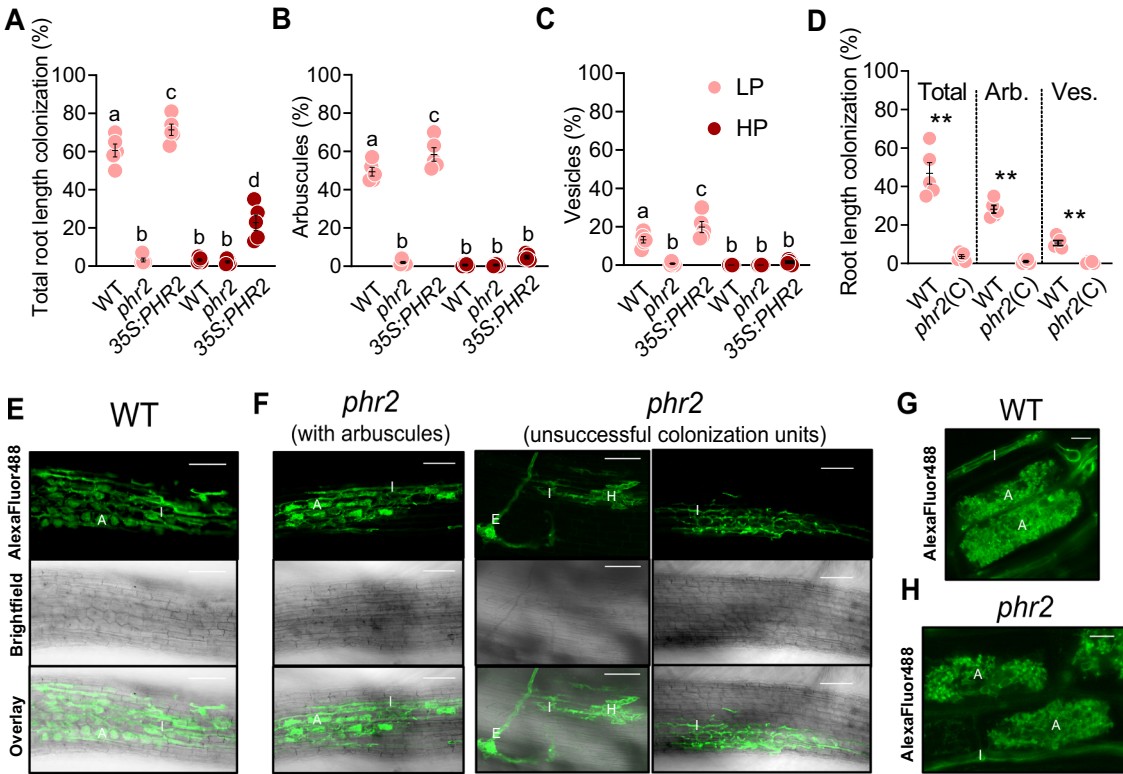

**Fig. 1 Effect of *PHR2* mutation and overexpression on root colonization by AM fungi.** Percent root length colonization (RLC) total (**A**), arbuscules (**B**), and vesicles (**C**) of the indicated genotypes inoculated with *R. irregularis* (AMF) for 7 weeks at low phosphate (25 µM $P_i$, LP) or high phosphate (500 µM $P_i$, HP). **D** Percent RLC of the indicated genotypes inoculated with AMF for 7 weeks at LP. Arb.: Arbuscules; Ves.: Vesicles. Confocal images of AMF in roots of wild type (**E**) and *phr2* (**F**). Arbuscule phenotype in wild type (**G**) and *phr2* (**H**) roots. E, extraradical hypha; H, hyphopodium; I, intraradical hypha; A, arbuscule. Scale-bars: E, F: 100 µm and G, H: 10 µm. Statistics: Individual data points and mean ± SE are shown. **A**–**C** N = 5 independent plants; Brown-Forsythe and Welch's One-Way ANOVA test with Games–Howell's multiple comparisons test. Different letters indicate statistical differences. **D** N = 5 independent plants; Mann–Whitney test (two-tailed) between *phr2*(C) and wild type for Total (p = 0.0079), arbuscules (p = 0.0079) and vesicles (p = 0.0079). Asterisks indicate significance of difference: ** p ≤ 0.01. **E**–**H** The phenotype was observed in 11 (6 + 5) independent plants in two independent experiments.

We compiled a list of 205 genes, called 'AM genelist' (Supplementary Data 4), containing genes with a genetically determined or putative function in AM, based on phylogenomic and/or conserved expression patterns across AM-competent plant species[23]. Almost 70% of genes in the AM genelist were less expressed in *phr2* vs. wild type, demonstrating that PHR2 is an important regulator of AM (Fig. 2G). Consistent with the low colonization of *phr2* and the presence of PHR binding sites in the promoters of some of the orthologs in *Lotus japonicus*[11], the activation of AM-induced genes was impaired, including a number of genes encoding (receptor) kinases, transcription factors, transporters, and enzymes with genetically determined function in AM (Supplementary Figs. 6–9 and Supplementary Data 5–7). Furthermore, many of these genes showed reduced expression in *phr2* even in the absence of the fungus (Supplementary Fig. 7). qPCR recapitulated this expression pattern of a selected subset of important AM-relevant genes (Supplementary Fig. 9). Some of these were activated by *35S:PHR2* at HP and most of them at MP, consistent with the degrees of promotion of root colonization by *35S:PHR2* at these non-permissive conditions (Fig. 1A–D and Supplementary Figs. 2, 3, 9, 10). Furthermore, 213 DEGs that were less expressed in *phr2* roots show increased expression in non-colonized *smax1* or *d3 smax1* roots, which display the opposite AM phenotype to *phr2*, i.e., they are colonized more extensively than wild type[24], and 41 of these genes are additionally shared with the AM genelist (Fig. 2G and Supplementary Data 7). SMAX1 is a

negative regulator of karrikin signaling and needs to be removed through proteasomal degradation after binding to a complex of the karrikin receptor KAI2 and the SCF$^{MAX2}$- complex to enable AM in rice[24,25]. Thus, karrikin signaling and PHR2 seem to target partially overlapping genes for regulating AM.

**PHR2 targets AM-relevant genes**. To understand whether PHR2 directly targets chromatin regions containing AM-relevant genes, we performed chromatin immunoprecipitation followed by DNA sequencing (ChIP-Seq). Since we surprisingly found that PHR2 promotes the expression of AM-relevant genes already in the absence of AMF at LP, we chose to perform ChIP-Seq under this condition. For the two independent replicates, more than 60% of PHR2 binding sites were found upstream of ATG (Supplementary Fig. 11). Motif enrichment analysis of DNA sequences associated with PHR2 binding sites revealed the P1BS element (GNATATNC) among the top 2 enriched motifs in both ChIP replicates (Supplementary Data 8 and Supplementary Fig. 12). Peaks representing PHR2 binding sites were annotated to nearby genes and 435 genes were shared between the two replicates (referred to as "PHR2 targets"; Supplementary Fig. 13A and Supplementary Data 8). 77% of these PHR2 targets with reduced expression in *phr2* contained P1BS/P1BS-like motifs in the sequence between the transcriptional start site (TSS) and 3000 bp upstream of the TSS (Supplementary Fig. 13B and Supplementary Data 9). ChIP-Seq quality was further validated by strong

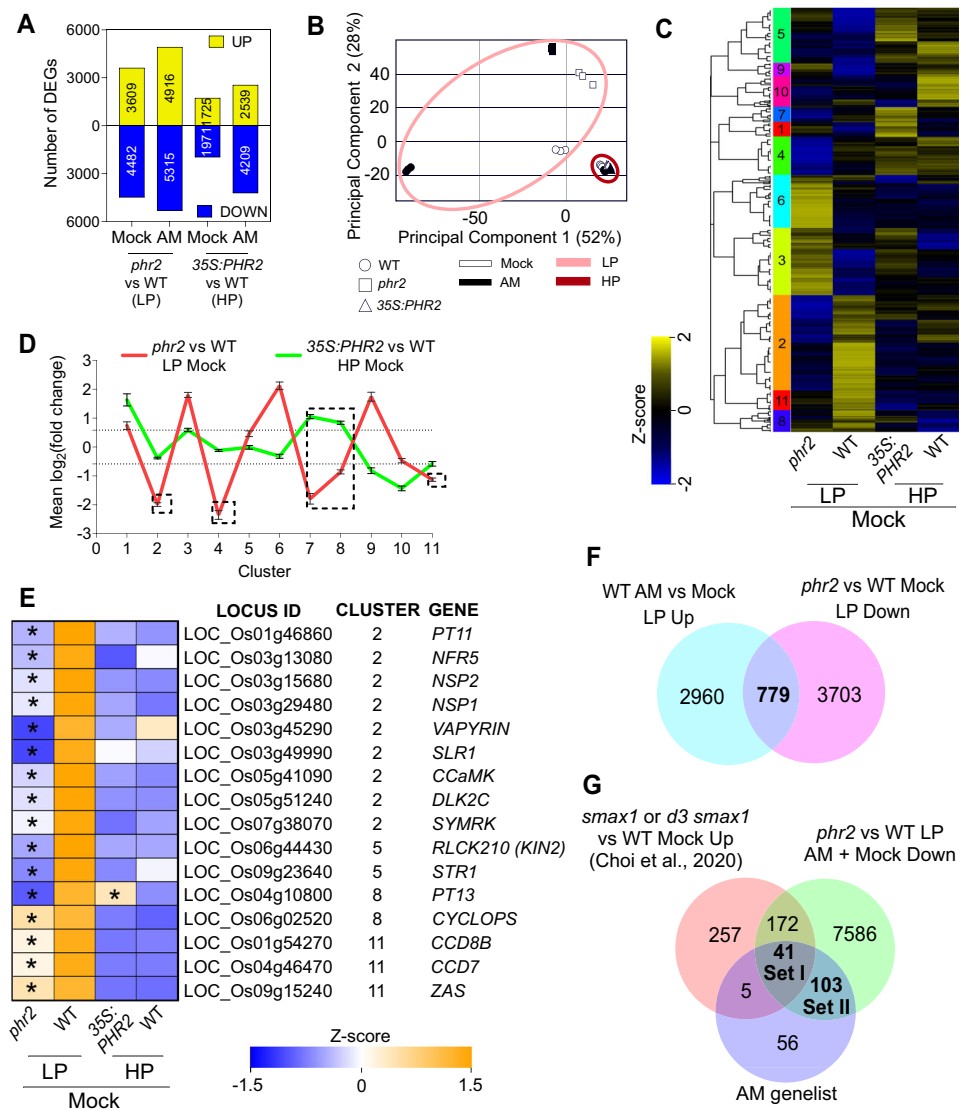

**Fig. 2 The PHR2-dependent transcriptome of non-colonized roots contains genes required for AM development. A** Number of up- and downregulated differentially expressed genes (DEGs) in *phr2* vs WT (LP) and *35S:PHR2* vs WT (HP) for mock or *R. irregularis*-inoculated plants. **B** PCA plot for the transcriptome of indicated samples. **C** Hierarchical clustering of combined DEGs from mock-inoculated roots using normalized counts. Colored bars on the left of the heatmap depict individual clusters (based on the dendrogram). Z-scores represent scaled normalized counts. **D** Mean $\log_2$FC of *phr2* vs WT (LP) and *35S:PHR2* vs WT (HP) for clusters obtained in (**C**). Dotted lines indicate mean $\log_2$FC values of −0.585 and 0.585 (used as a cut-off for selecting DEGs). Dashed boxes highlight the mean $\log_2$FC of clusters with overall reduced transcript accumulation in *phr2* vs WT. **E** Z-scores for a subset of genes from (**C**) previously reported being functionally required in AM. Asterisks indicate that the gene is a DEG in the *phr2* vs WT (LP) or *35S:PHR2* vs WT (HP) comparisons. **F** Venn diagram showing overlap of DEGs with increased transcript accumulation in wild type 'AM vs Mock' and DEGs with reduced transcript accumulation in '*phr2* vs WT' in Mock at LP. **G** Venn diagram showing the intersection of DEGs with reduced transcript accumulation in the '*phr2* vs WT' comparison for mock and AMF-inoculated samples with AM genelist (Supplementary Data 4) and DEGs upregulated in either *smax1* (vs WT) or *d3 smax1* (vs WT) Mock[24]. Statistics: Mean ± SE are shown in (**D**) where N = Number of genes in each cluster as obtained in (**C**). Error bars represent the SE of the mean.

PHR2-binding peaks at P1BS elements in the promoter regions of the previously known PHR2-targets, *GDPD2, IPS1,* and *SPX1/2/3*[17,26,27] (Supplementary Fig. 14 and Supplementary Data 10). An overlap of PHR2 targets with DEGs reduced in *phr2* vs wild type resulted in 162 overlapping genes, for which GO enrichment analysis revealed that most of these were involved in "Cellular response to phosphate starvation" and motif enrichment in the 3 kb region upstream of TSS revealed P1BS motif (GNATATNC) as the topmost hit (Fig. 3A, B and Supplementary Fig. 13C, D). 17 genes overlapped among the two ChIP replicates and the AM genelist, and these represent a significant enrichment of AM-relevant genes in high-confidence PHR2 targets, when compared

to the whole rice genome (*p*-value of $6.4 \times 10^{-11}$) (Supplementary Fig. 15). An overlap also showed 27 common genes (including genes from individual ChIP replicates), 15 of which were repressed in *phr2* vs wild type and common to both ChIP replicates (Fig. 3A, C and Supplementary Data 10). These 15 genes contain 8 with established and crucial roles in AM. Except for *PT11*, which is required for mycorrhizal phosphate uptake and arbuscule maintenance, all these genes are indispensable for pre-contact signaling and necessary for or involved in important early events of fungal root entry (Fig. 3C), congruous with the *phr2* AM phenotype. The remaining 12 genes that were hit in only one ChIP-Seq replicate included the genes *CYCLOPS*,

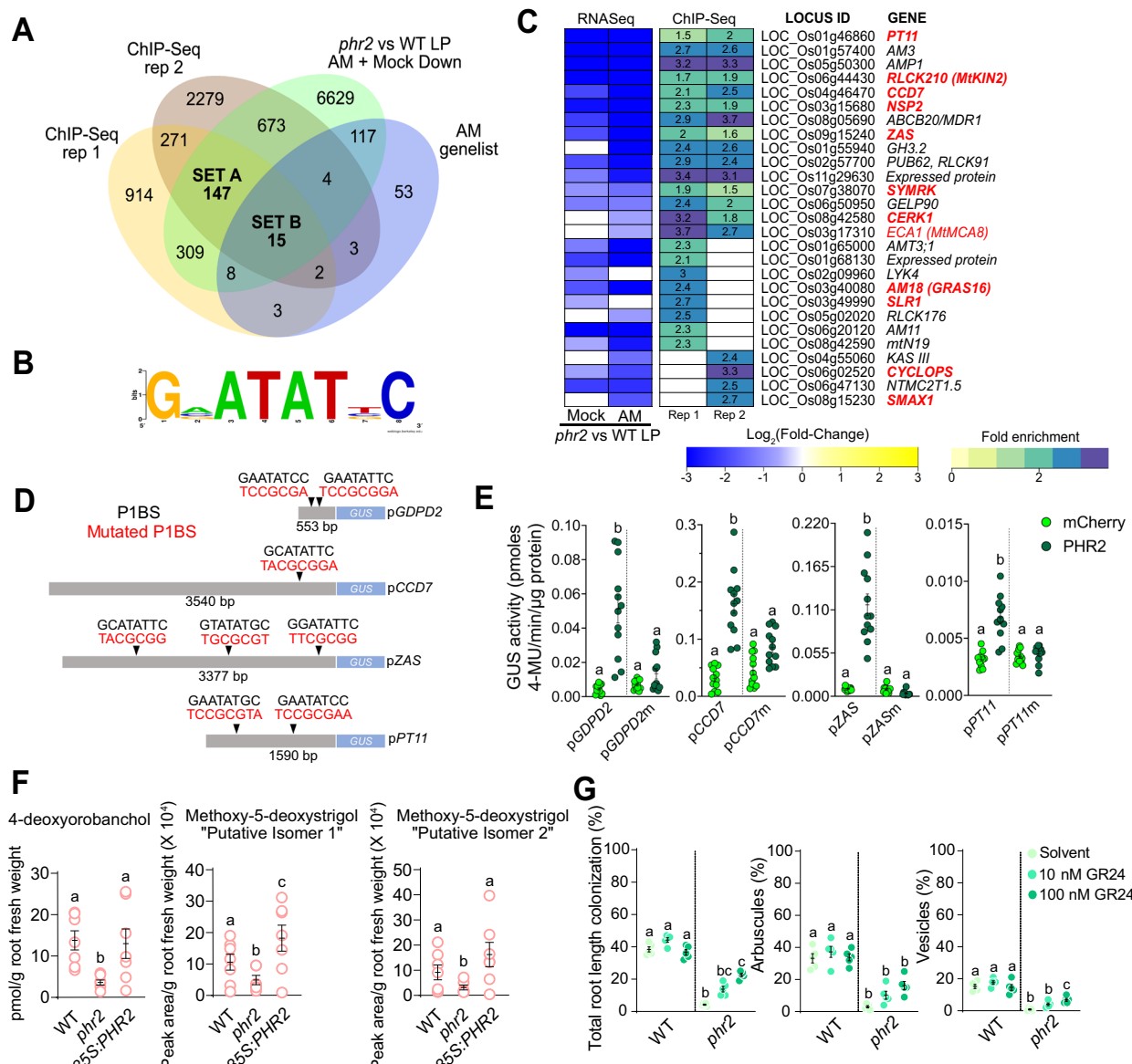

**Fig. 3 PHR2 targets important AM symbiosis genes. A** Overlap of PHR2 ChIP-Seq targets with RNASeq DEGs displaying reduced expression in Mock and AM roots of *phr2* vs WT roots grown at LP and with AM genelist (Supplementary Data 4). **B** Consensus PHR2-binding motif for Set A + Set B genes based on 'RSAT Plants' oligo-analysis. **C** Fold change of RNAseq-based transcript accumulation in Mock and AM roots of *phr2* vs WT grown at LP (left) and ChIP-Seq fold enrichment for PHR2 binding (right) for the 27 genes hit in at least one ChIP-Seq replicate and overlapping with *phr2* vs WT down and AM genelist. Genes with assigned functional roles in AM, based on mutant phenotypes are displayed in red (bold for mutants, regular font for RNAi lines). **D** Illustration of promoters used in transactivation assays with positions of PHR1 binding site (P1BS) elements (black) and mutated P1BS elements (red). **E** PHR2 transactivates the promoters of *GDPD2, CCD7, ZAS,* and *PT11* (553, 3540, 3377, and 1590 bp upstream of ATG, respectively) in a *P1BS* element-dependent manner in *Nicotiana benthamiana* leaves. *promoter:GUS* fusions were co-transformed with *35S:PHR2* or *35S:mCherry* (negative control). 'm' indicates that all *P1BS* elements in the promoter were mutated. Individual dots show GUS activity in protein extracts from leaf disks from four independent plants. GUS: β-glucuronidase. **F** Quantity of 4-deoxyorobanchol and two isomers of methoxy-5-deoxystrigol in the root exudates of WT, *phr2*, and *35S:PHR2* grown at LP. **G** Percent root length colonization total (left), arbuscules (middle), and vesicles (right) of the indicated genotypes inoculated with *R. irregularis* for 6 weeks at low phosphate (25 μM $P_i$) and treated with 0.02% acetone solvent or the synthetic SL analog *rac*-GR24. Statistics: Individual data points and mean ± SE are shown. **E** $N = 11$–12 biological replicates representing independent *Agrobacterium* infiltrations, one per leaf into 3 leaves of 4 individual plants; Kruskal–Wallis test with Dunn's posthoc comparison. **F** $N = 6$-7 independent plants; Brown–Forsythe and Welch's One-Way ANOVA test with Games–Howell's multiple comparisons test. **G** $N = 5$ independent plants; Brown–Forsythe and Welch's One-Way ANOVA test with Games–Howell's multiple comparisons test.

*SLR1/DELLA*, required for root colonization and arbuscule formation[22] (Fig. 3C) as well as *GRAS16* and *SMAX1*, which are positive and negative regulators of AM respectively[24,28].

Specific binding of PHR2 to the P1BS/P1BS-like motif in the promoters of the 8 AM-relevant PHR2 targets detected in both ChIP-Seq replicates was confirmed by ChIP-qPCR except for

*CERK1* (Supplementary Figs. 16 and 17). Surprisingly, for the *CERK1* promoter, we found moderate enrichment at a CTTC/MYCS motif nearby the P1BS motif. It is possible that a CTTC-binding protein (e. g. a WRI transcription factor[12]) forms a complex with PHR2 to regulate *CERK1* transcription and that this CTTC/MYCS binding factor has a stronger affinity to the

DNA than PHR2, leading to enrichment of this motif after PHR2 pulldown. Using transactivation assays in *Nicotiana benthamiana* leaves, we found that the promoter of the established PHR2 target *GDPD2* and three previously uncharacterized AM-relevant PHR2 target promoters p*CCD7*, p*ZAS*, and p*PT11* are transactivated by PHR2 in a P1BS-dependent manner (Fig. 3D, E), confirming the ChIP-Seq results.

The binding of and transactivation of the promoter of the strigolactone (SL)-biosynthesis gene *CCD7* by PHR2 (Fig. 3C–E) and the decreased expression of *CCD7* and *CCD8B* in the *phr2* mutant (Fig. 2E and Supplementary Figs. 9 and 10) are consistent with the observation that strigolactone biosynthesis is increased at $P_i$ starvation[29]. Exuded SL in the rhizosphere activates AMF and is crucial for relevant levels of root colonization, while root colonization of *ccd7* and *ccd8* mutants with defects in SL biosynthesis is strongly reduced[30–33]. To understand the impact of mutating *phr2* on SL exudation we quantified the canonical rice SL 4-deoxyorobanchol as well as 2 isomers of Methoxy-5-deoxystrigol in the root exudates of wild type, *phr2*, and *35S:PHR2* (Fig. 3F). The amount of all three compounds was reduced by at least half in *phr2* root exudates as compared to the wild type, while the Methoxy-5-deoxystrigol isomers were slightly increased in the exudates of *35S:PHR2*. This indicates that PHR2 indeed regulates SL biosynthesis and thereby the amount of SL in the root exudate. We examined whether exogenous supplementation of *phr2* with SL can restore AM colonization. Treatment of roots with 100 nM of the synthetic SL analog, *rac*-GR24 led to a significant increase in total colonization and vesicles in *phr2* roots but not in wild type roots, which probably exude saturating amounts of SL (Fig. 3G). Thus, part of the inability of *phr2* to support AM is due to reduced SL exudation, while another part can be explained by reduced expression of other genes required for root colonization and AM functioning (Fig. 2E).

**Transcript accumulation of gibberellin-related genes is altered in *phr2*.** Gibberellin (GA) treatment inhibits AM development because AM formation requires DELLA, the proteolytic target of GA-signaling[34,35]. It was recently shown that GA is also involved in the suppression of AM in response to HP, as GA-deficient *Nicotiana tabacum* plants were less sensitive to AM inhibition by HP than wild type[36]. To understand whether GA signaling participates in the regulation of AM in response to phosphate downstream of PHR2, we mined the RNAseq and ChIP-Seq data for GA-biosynthesis and signaling genes. In addition to *DELLA* and the *GIBBERELLIN INSENSITIVE DWARF 2* (*GID2*), encoding an F-box protein required for GA-perception being ChIP targets of PHR2 (Fig. 3C and Supplementary Fig. 18), we found that GA-related genes were enriched in genes deregulated in *phr2* (Supplementary Fig. 18). GA biosynthesis genes have lower expression levels in Mock roots of *phr2* mutants as compared to wild type, while the transcript levels of three genes encoding enzymes involved in de-activating GA (*GA13ox2*, *GA2ox3*, and *GA2ox7*) are increased in *phr2* (Supplementary Fig. 18). Thus, the expression of several genes involved in GA metabolism is controlled by PHR2.

**The role of PHR in regulating AM is conserved in the dicotyledon *Lotus japonicus*.** We used the model legume *Lotus japonicus* to understand whether the role of PHR in AM is conserved in dicotyledons. The *L. japonicus* genome contains three *PHR* genes, which we named *PHR1A*, *PHR1B*, and *PHR1C* (Supplementary Fig. 19). A LORE1 retrotransposon insertion line[37] was obtained for *PHR1A* (Supplementary Fig. 19). Mutation of *Ljphr1a* was sufficient to cause a significant reduction in colonization by *R. irregularis* as compared to the wild type of

hairy roots transformed with an empty vector and grown at LP. This indicates that the role of PHR in AM is conserved in *L. japonicus*. However, there may be redundancy among the three PHR proteins, and the AM phenotype may be stronger in double and triple *phr* mutants. Transgenic expression of *PHR1A* driven by a *Ubiquitin* promoter in *phr1a* hairy roots restored colonization to wild-type levels. Furthermore, *Ubiquitin* promoter-driven expression of *PHR1A* significantly increased root colonization at high phosphate as compared to the wild type (Fig. 4A). We examined, whether mutation of *PHR1A* leads to reduced expression of AM-relevant genes in *L. japonicus*, and monitored transcripts of selected candidate genes in Mock roots by RT-qPCR, namely the common symbiosis genes *SYMRK*, *CCaMK*, and *CYCLOPS* and the three SL biosynthesis genes *D27*, *CCD7*, and *CCD8*. Similar to rice *phr2* mutants, the expression of *SYMRK*, *CCaMK* and *CYCLOPS* was reduced in roots of *phr1a*. However, no significant reduction in transcript accumulation of the SL biosynthesis genes was observed, even though one P1BS is located in the promoter of each of these genes (Supplementary Fig. 20). It is possible that one of the other PHRs regulates their expression. Alternatively, SL biosynthesis genes may not be regulated by PHRs in *Lotus japonicus*.

**PHR2 promotes AM and yield in field soil**. To assess the importance of rice *PHR2* in supporting AM in field soil and in promoting plant performance via AM we conducted a greenhouse experiment with soil from a rice paddy field with (HP) or without (LP) $P_i$ fertilization (Fig. 5 and Supplementary Figs. 21–25). At LP, inoculation with *R. irregularis* caused a significant increase in several important agronomic parameters: total shoot phosphorus (P) and shoot P concentration, root fresh weight, and most importantly seed setting and 1000 grain weight in wild type and *35S:PHR2*. However, these parameters remained unchanged or decreased in *phr2* at LP (Fig. 5 and Supplementary Fig. 22), showing that *phr2* is required for AM-mediated yield increase in field soil. At HP, total shoot phosphorus (P), root fresh weight, seed setting, and 1000 grain weight remained the same in wild type independent of AMF presence. However, in *35S:PHR2* these parameters increased in response to AM. Furthermore, *35S:PHR2* caused a significant response to AM for plant height, panicle length, and shoot dry weight at LP, HP, or both (Supplementary Fig. 22). Consistently, *35S:PHR2* enabled AM colonization in field soil even at HP, accompanied by increased expression of the AM marker genes *AM1*, *AM3*, *AM14/ARK1*, and *PT11*[14] (Supplementary Figs. 23–25).

Our data confirm previous observations that *35S:PHR2* reduces plant growth as compared to wild type (Fig. 5 and Supplementary Fig. 22). This may be due to $P_i$ toxicity;[15] changes in hormone biosynthesis and/or signaling or other pathways affecting development. Interestingly, AM mitigates this negative effect on rice performance, perhaps because it promotes the uptake of other nutrients, which consequently become less limiting and/or affects other pathways regulating plant growth.

**Discussion**
Here we demonstrate that rice PHR2 is essential for the establishment of AM symbiosis under laboratory conditions, as well as for AM-mediated improvement of rice yield in LP paddy field soil.

PHR2 supports AM development at LP because it promotes the expression of a number of genes required for AM development and function already in the absence of AMF. This possibly occurs through the effect of PHR2 on chromatin accessibility, as recently shown in *Arabidopsis*[38]. We detected PHR2 targets using ChIP-Seq in the absence of AMF and these comprise

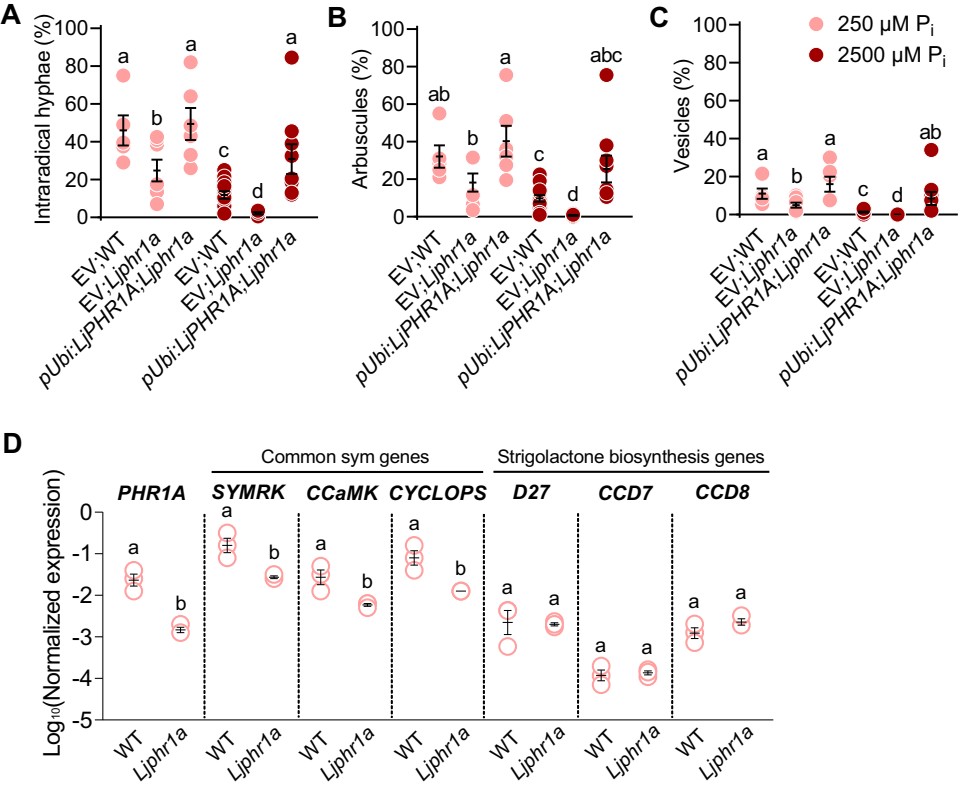

**Fig. 4 PHR1A is required for full colonization of *Lotus japonicus* roots by *R. irregularis*.** Percent intraradical hyphae (**A**), arbuscules (**B**), and vesicles (**C**) of hairy roots for indicated genotypes inoculated with *R. irregularis* (AMF) for 4 weeks at 250 μM $P_i$ or 2500 μM $P_i$. EV, empty vector. **D** Relative transcript accumulation in mock-inoculated (Mock) hairy roots of the indicated genotypes in parallel with the experiment in Fig. 5A–C. Statistics: Individual data points and mean ± SE are shown. **A–C** $N$ = 5–13 independent plants; Brown–Forsythe and Welch's One-Way ANOVA test with Games–Howell's multiple comparisons test. Different letters indicate statistical differences between the samples. **D** $N$ = 3 independent root systems; Mann–Whitney test (two-tailed). Different letters indicate statistical differences between statistical groups.

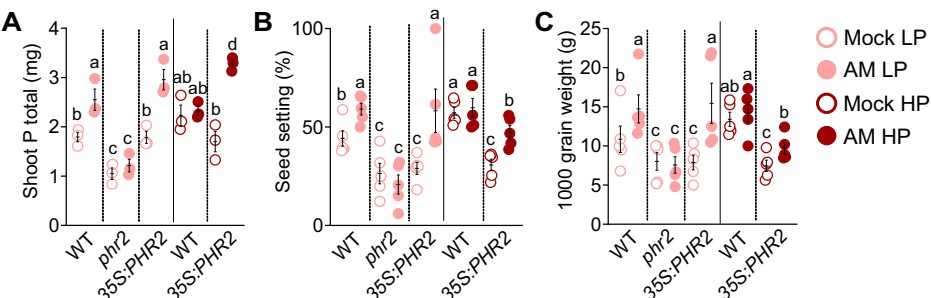

**Fig. 5 PHR2 affects AM-mediated phosphate uptake and yield in field soil. A** Total shoot phosphorus (mg), **B** seed setting, and **C** 1000 grain weight of the indicated genotypes inoculated with *R. irregularis* (AM) or non-inoculated (Mock) and grown at LP (unfertilized) or HP (fertilized with superphosphate fertilizer, $P_2O_5$). Plants were grown in a greenhouse in soil from the Longhua field base in Shenzhen, China, and harvested at 110 days post transplanting (dpt) for trait quantification. Statistics: Individual data-points and mean ± SE are shown. $N$ = 3–5 independent plants; Brown–forsythe and Welch's One-Way ANOVA test with Games–Howell's multiple comparison test was carried out. Different letters indicate statistical differences between genotypes and treatments.

strigolactone biosynthesis genes needed for strigolactone exudation to activate the fungus[31,33], membrane receptors needed for perception of fungal signals[39,40] and genes belonging to the common symbiosis signal transduction cascade, which is thought to relay the signals to the nucleus to activate gene expression, and which is required for fungal entry into the root[14]. Furthermore, PHR2 activates genes encoding transporters needed for nutrient exchange at the peri-arbuscular membrane, such as *PT11*[41]. Thus, PHR2 appears to be required for the activation of genes involved in several steps of arbuscular mycorrhiza development

from initiation of root colonization to its functioning in nutrient exchange (Fig. 6).

Shi et al. (2021)[18] performed a large yeast-1-hybrid screen using 47 promoters of genes activated by AM predominantly in arbuscule-containing cells. They found that a large proportion of them was bound by PHR2, consistent with the previous observation that a number of their orthologs in *L. japonicus* carry the P1BS element in their promoter[11]. Interestingly, although transcripts of these genes were reduced in *phr2* mutants in our study, which can be a symptom of reduced root colonization, we did not

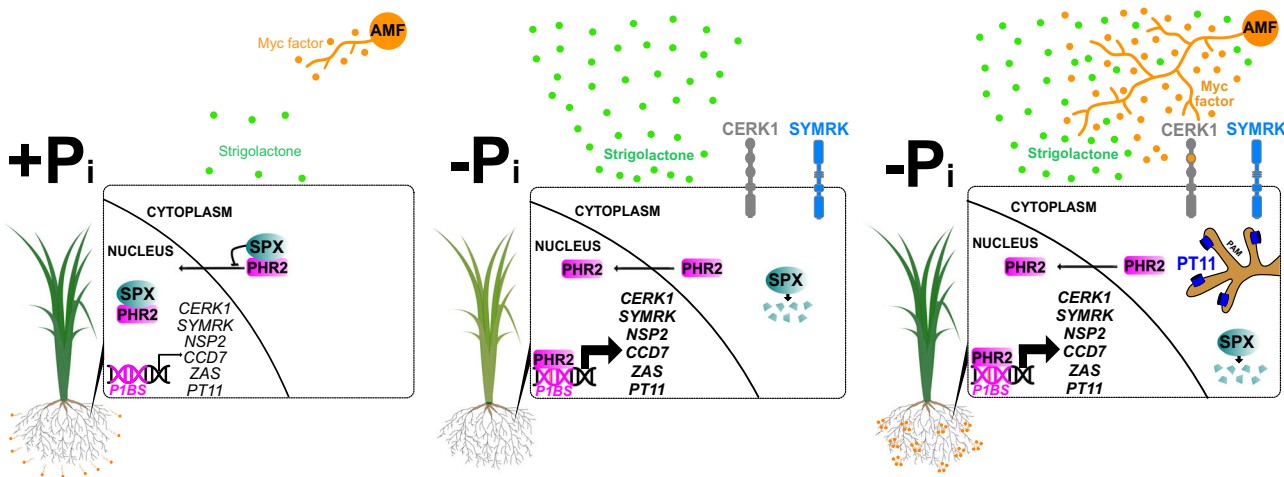

**Fig. 6 Model depicting regulation of AM symbiosis by PHR2.** When plants obtain sufficient phosphate (left), SPX proteins prevent nuclear translocation of PHR2[42], as well as PHR2 binding to promoters of phosphate starvation -induced genes including AM relevant genes[17,42]. This causes low exudation of strigolactone and poor expression of genes required for perception of Myc-Factors and fungal entry, thereby preventing full symbiosis development. Upon phosphate starvation, SPX proteins, are degraded[42]. Consequently, PHR2 is active, can bind to P1BS elements in promoters, and transcriptionally activate genes important for AM, such as *CCD7* involved in strigolactone biosynthesis for activation of the fungus in the rhizosphere prior to contact[31–33], genes encoding receptors involved in the perception of fungal signals prior to root contact such as *CERK1* and *SYMRK*[39,40,65,66], the transcription factor *NSP2*[67], *ZAS* involved in apocarotenoid biosynthesis promoting root colonization[68], and the AM-specific phosphate transporter gene *PT11*[41] (localized to the peri-arbuscular membrane (PAM)) required for Pi uptake from the fungus. (For simplicity, we focus here on 6 genes with important and genetically determined roles in AM that have been recovered in both ChIP-Seq replicates Fig. 3C, and were confirmed by ChIP-qPCR, Supplementary Fig. 17). Consequently, at low phosphate, roots exude increased amounts of strigolactone and can perceive fungal signals (middle), the fungus is activated to colonize the roots and the symbiosis can function through nutrient transporters localizing to the peri-arbuscular membrane (right). Thus, symbiosis establishment appears to be enabled as a part of the PHR2-regulated phosphate starvation response.

capture most of these genes in our ChIP-Seq experiment except for the AM-specific transporter genes *PT11* and *AMT3.1*. We conducted ChIP-Seq from non-colonized roots and it is possible that additional AM-induced transcription factors are necessary to stabilize PHR2-binding to the promoters of these AM-induced genes. However, our ChIP-Seq approach enabled us to identify unexpected PHR2 targets involved in pre-contact symbiotic signaling and early stages of symbiosis, which were not present in the pre-selected set of promoters analyzed by Y1H[18].

At HP the movement of PHR2 to the nucleus and its binding to promoters is inhibited by SPX proteins, which act as phosphate sensors[17,18,42,43]. Consistently, it appears that AM development is attenuated at HP due to SPX-mediated inhibition of PHR2-mediated transcriptional activation of AM-relevant genes. This is consistent with the finding that ectopic expression of PHR2, driven by a strong constitutive promoter allows AM development at HP, likely because PHR2 outnumbers SPX proteins when it is overexpressed. Importantly, a recent study[18] showed that quadruple mutants in rice *SPX* genes have significantly increased root colonization independently of the phosphate concentration in the fertilizer, while ectopic SPX expression leads to significantly reduced root colonization. In contrast, *Medicago truncatula* SPX1 and SPX3 were shown to positively affect root length colonization by promoting the expression of SL biosynthesis genes. However, they also seemed to promote arbuscule degeneration and turnover, indicating a dual role of these proteins in regulating AM in *M. truncatula*[44]. The contrasting findings in rice and *M. truncatula* raise the possibility that different SPX family members play divergent roles in AM or that they differently affect AM in grasses vs legumes.

We show that the role of PHRs in AM is conserved in the model legume *L. japonicus*. It is therefore likely, that AM symbiosis is a part of the PHR-mediated phosphate starvation response syndrome at least across the angiosperms. It will be interesting to learn whether this hypothesis can be confirmed in

other clades of the vascular and early-diverging non-vascular plants. In summary, we provide important insights into the regulation of AM by the plant phosphate status, which has a broad significance in agriculture and terrestrial ecosystems.

## Methods

### Plant material, growth conditions, and plant co-cultivation with AM fungi.
*Oryza sativa* L. ssp. *japonica* genotypes used in this work were wild type (cv. ZH11) and CRISPR-Cas9 generated *phr2*(C)[19] and wild type (cv. Nipponbare), *phr2*, *35S:PHR2*, and *35S:PHR2-FLAG*[17]. Rice seeds were sterilized by incubating in absolute ethanol for 2 min, then in sterilization solution (10% Clorox + 0.1% SDS) for 30 min and subsequently in autoclaved distilled water for 1 h. Seeds were germinated on agar plates and kept in the dark for 3 days at 27 °C and 60% air humidity in a controlled plant growth chamber. Subsequently, germinated seedlings were exposed to light and grown for 2 weeks in a long-day photoperiod (16 h light/8 h dark) in the same temperature and humidity conditions. 2-week-old seedlings were then transplanted to pots filled with quartz-sand and inoculated with 500 spores of *Rhizophagus irregularis* DAOM 197198 (Agronutrition, Toulouse, France) per plant and grown at the same conditions. Plants were fertilized with 50 ml sterilized half-Hoagland media at 25 μM Pi (low phosphate, LP), 200 μM Pi (medium phosphate, MP), and 500 μM Pi (high phosphate, HP) concentrations once a week and watered a second time per week with autoclaved distilled water.

For *L. japonicus*, ecotype Gifu wild type and *phr1a* mutants were used. The *phr1a* mutant corresponds to the LORE1 insertion line 30063054[45]. Seeds were scarified with sandpaper and surface sterilized with 0.1% SDS in 2% NaClO, washed 6 times with sterile water, and left to swell in sterile water with permanent inversion for 2 h. The seeds were germinated on 0.8% B5-Agar at 22 °C in darkness for 4 days and then in light for 3 days. Plates were then transferred to closed, sterilized cultivation pots (Duchefa, Os 140 Box, Green Filter), containing 450 g of sand and watered with modified B&D medium, containing 250 μM Pi (low phosphate) or 2.5 mM Pi (high phosphate)[46]. For arbuscular mycorrhiza colonization assays, plant roots were inoculated with 500 spores per plant of *R. irregularis* DAOM197198 (Agronutrition, Toulouse, France). The plants were grown for 4 weeks in a chamber set at 22 °C in a long-day photoperiod (16 h light/8 h dark) with 60% relative humidity.

### Quantification of root colonization and AM imaging.
Rice roots were harvested at 7 weeks post-inoculation (wpi) into 10% KOH and boiled for 15 min followed by incubation in 10% acetic acid, ink staining, and 5% acetic acid de-staining as described[46]. Root pieces of 1 cm each were then mounted on a slide for observation

under an inverted light microscope to quantify total AM colonization and % of AM structures using the gridline intersect method[47]. Furthermore, images of these stained roots were visualized using an OPTO EDU Binocular Compound Microscope (Opto-Edu, Beijing, China) fitted with 5MP USB 2.0 Color CMOS Digital Eyepiece Microscope Camera (AmScope, USA) and exported with Swift Easy View software (Motic, China). For visualization of AM structures inside root cells, roots were stained with Wheat Germ Agglutinin, Alexa Fluor™ 488 Conjugate[46]. Roots were then visualized with a Zeiss LSM 710 Confocal Microscope (Zeiss, Germany) at 20X for a root overview or at 100X (oil) for details of arbuscules and images extracted using the ZEN 2020 v3.3 software (Zeiss, Germany).

*L. japonicus* roots were stained with acid ink[47]. Root length colonization was quantified using a modified gridline intersect method[48] at 10× magnification under a light microscope Leica, type 020-518500 DM/LS (Leica, Germany).

**RNA extraction and RT-qPCR**. Plant roots were harvested, frozen in liquid nitrogen, and stored at −80 °C. Frozen roots were ground into powder using mortar and pestle and the powder was immediately transferred to lysis extraction buffer and processed for RNA isolation using FastPure Plant Total RNA Isolation Kit (Polysaccharides & Polyphenolics–rich) (Vazyme, China). Isolated RNA was checked for the absence of genomic DNA contamination by PCR. RNA was quantified with a Nanodrop (Thermo Scientific, USA). First-strand cDNA synthesis was carried out with HiScript® III RT SuperMix for qPCR (+gDNA wiper) (Vazyme, China). qPCR was conducted with primers indicated in Supplementary Table 1 and ChamQ Universal SYBR® qPCR Master Mix (Vazyme, China) on a Bio-Rad CFX384 qPCR cycler (Bio-Rad Laboratories, Shanghai, China). Threshold cycle values were extracted using the CFX Maestro Software 2.0 (BioRad, USA).

RNA from *Lotus japonicus* roots was extracted using the Spectrum Plant Total RNA Kit (Sigma, USA). DNA contamination was removed by Invitrogen DNAse I amp. grade (www.invitrogen.com) and RNA was tested for genomic DNA contamination by PCR. cDNA synthesis was performed with 600 ng RNA using the iScript cDNA synthesis kit (Bio-Rad, Germany). Real-time RT-PCR was performed with the qPCR GreenMaster high ROX mix (Jena Bioscience, Germany) and primers shown in Supplementary Table 1. The qPCR reaction was run on an iCycler (Bio-Rad, Germany) according to the manufacturer's instructions. Thermal cycler conditions were: 95 °C 2 min, 45 cycles of 95 °C 30 s, 60 °C 30 s and 72 °C 20 s followed by dissociation curve analysis. Expression levels were calculated according to the ΔΔCt method[49]. For each genotype and treatment, three biological replicates were monitored and each sample was represented by two to three technical replicates.

**RNASeq and data analysis**. RNA was extracted as described above, checked for quality using an Agilent Bioanalyzer (Agilent, USA), and quantified in a Qubit 3.0 fluorometer (Invitrogen, USA). Samples with a RIN value of > 7 were used for library preparation (Illumina TruSeq Stranded Total RNA kit, Illumina, USA) and sequenced on an Illumina HiSeq 4000 to obtain 150 bp paired-end reads. Raw fastq files were processed through adaptor-trimming. Clean reads were tested with FastQC (https://www.bioinformatics.babraham.ac.uk/projects/fastqc/) for base quality. Data were processed with a quasi-transcript mapping approach in SALMON[50]. SALMON 0.7.0 was operated in the Conda environment inside Linux (https://anaconda.org/bioconda/salmon). Reads were mapped onto the *Oryza sativa* reference transcriptome downloaded from Phytozome v12 (https://phytozome.jgi.doe.gov/pz/portal.html). Read counts were obtained for rice transcripts at the gene level. Read counts were further processed through tximport in R/Bioconductor 3.13 (https://cran.r-project.org/; R Core Team, 2013) for input into EdgeR for Data exploratory and differential expression analysis. Differentially expressed genes (DEGs) were obtained with the cut-off: absolute(log₂Fold-Change) ≥ 0.5845 and FDR ≤ 0.05. Heatmaps were prepared using the heatmap.2 package in R/Bioconductor. AgriGO was utilized to find enriched GO terms for DEGs[51]. A list of genes implicated in AM was compiled based on previous studies in rice and other AM host plants and is referred to as "AM genelist" (Supplementary Data 4). For genes identified to have a role in AM development of symbiosis in other AM host plants, reciprocal blast was used to identify the best match in rice.

**Chromatin immunoprecipitation (ChIP) and sequencing**. Chromatin immunoprecipitation was performed as described[52] with few modifications as follows: 3–4-week-old *35S-PHR2-FLAG* transgenic rice roots were ground to a fine powder in liquid nitrogen and stored at −80 °C until further use. Subsequently, 3 g of tissue powder were resuspended in cold extraction buffer with 1.5% formaldehyde and incubated at room temperature for 10 min with shaking, followed by quenching for 5 min with 125 mM glycine and 750 mM Tris's buffer. After Miracloth filtration and subsequent nuclei enrichment and lysis steps, the resulting chromatin was sheared in Bioruptor® Pico sonication device (Diagenode, USA) with settings of 30 s ON/30 s OFF for 40 cycles. Agarose gel check on sonicated chromatin (reverse cross-linked and DNA purified) suggested that most fragments ranged from 150 to 500 bp. Sonicated chromatin was concentrated using Microcon® Centrifugal Filters (Merck Millipore, USA). After centrifugation, sonicated chromatin supernatant was selected for immunoprecipitation. 20 μl chromatin was saved at −20 °C for input DNA, and 100 μl chromatin was used for immunoprecipitation by anti-Flag antibodies (F1804, Sigma) and IgG antibodies (ab171870, Abcam) respectively. The

amount of each antibody used in each sample was 10 μg and antibody dilution was 1: 100. Antibody was used in the immunoprecipitation reactions at 4 °C overnight. The next day, 30 μL of protein beads were added and the samples were further incubated for 3 h. Next, the beads were washed once with 20 mM Tris/HCL (pH 8.1), 50 mM NaCl, 2 mM EDTA, 1% Triton X-100, 0.1% SDS; twice with 10 mM Tris/HCL (pH 8.1), 250 mM LiCl, 1 mM EDTA, 1% NP-40, 1% deoxycholic acid; and twice with TE buffer 1× (10 mM Tris-Cl at pH 7.5 + 1 mM EDTA). Bound material was then eluted from the beads in 300 μL of elution buffer (100 mM NaHCO₃, 1% SDS), treated first with RNase A (final concentration 8 μg/mL) during 6 h at 65 °C and then with proteinase K (final concentration 345 μg/mL) overnight at 45 °C. To examine the quality of ChIP DNA to assess enrichment for PHR2 binding motif P1BS, a ChIP-qPCR was conducted, using immunoprecipitated (anti-Flag and IgG) DNA and input DNA and specific primers for sequence flanking the P1BS motif in the *IPS1* promoter. qPCR reactions were run using the ChamQ SYBR Color qPCR Master Mix (Vazyme Biotech, China) on Bio-RAD CFX384 real-time system (BioRad, USA). The enrichment values were normalized to the input sample. After normalizing with input, fold enrichment was calculated using the "ΔΔCₜ" method by comparing with IgG negative control immunoprecipitated sample. Only samples showing more than five-fold enrichment compared to IgG for the *IPS1* promoter[17] negative control were used for sequencing. For ChIP-Seq, immunoprecipitated and input DNA were used to construct sequencing libraries following the protocol provided by the NEXTflex® ChIP-Seq Library Prep Kit for Illumina® Sequencing (NOVA-5143, Bioo Scientific) and sequenced on HiSeq X Ten (Illumina, USA) with PE 150 method.

**ChIP-Seq data analysis**. Sequencing yielded approximately 20 million paired-end reads of 150 bp per sample. Quality control of raw reads was performed in FASTQC. Adaptor trimming was carried out with TRIM-Galore 0.6.4 (https://github.com/FelixKrueger/TrimGalore) followed by mapping to reference genome in BWA-MEM2[53]. MACS2 was used to identify narrow or broad peaks representing PHR2 binding sites (showing enrichment of FLAG-immunoprecipitated sequencing reads over input sequencing reads) from non-duplicated uniquely mapped reads[54]. Peaks were extracted with a threshold of 1.5-fold enrichment over INPUT and significance p-value ≤ 0.05. Fasta sequence for PHR2 binding sites and peak annotation to nearby genes were performed in samtools-1.14 and bedtools 2.29.2 respectively[55,56]. Read counts were normalized to CPM (Counts per Million mapped reads) and coverage was generated in deepTools 2 and visualized with IGV 2.8.9[57,58]. Motif enrichment analysis for PHR2 binding sites was carried out in STREME (https://meme-suite.org/meme/tools/streme). Motif enrichment analysis for Set A + Set B genes in Fig. 3A was carried out in RSAT-oligo-analysis (http://rsat.eead.csic.es/plants/oligo-analysis_form.cgi)[59]. GO enrichment on Set A + Set B genes was performed in AgriGO. The search for P1BS or P1BS-like motifs in the sequences 3000 bp regions (upstream of TSS) for PHR2 target genes identified in the two ChIP-Seq replicates separately and for genes with repression in *phr2*, was performed with RSAT-dna-pattern (http://rsat.eead.csic.es/plants/dna-pattern_form.cgi).

**ChIP-qPCR**. ChIP was performed as mentioned above. Input DNA and FLAG- and IgG-immunoprecipitated DNA from three independent ChIP experiments were used for qPCR. qPCR was conducted with primers indicated in Supplementary Table 2 and TransScript® Green qRT-PCR SuperMix (TransGen Biotech, Beijing, China) on a Bio-Rad CFX384 qPCR cycler (Bio-Rad Laboratories, Shanghai, China).

**rac-GR24 treatment**. Plants inoculated with 500 spores of *R. irregularis* as described above were treated with a synthetic analog of strigolactone, *rac*-GR24 (Chiralix, CX23880). *rac*-GR24 was dissolved at 10 mM concentration in 100% acetone to make stocks and directly added to the Hoagland medium for use at 10 and 100 nM concentrations. The control solution was supplied with equal amounts of acetone. Plants were watered 2 times per week for 6 weeks. At 6 weeks, roots were harvested for acid-ink staining and quantification of colonization.

**Plasmid construction**. Genes and promoter regions were amplified with NEB Q5 polymerase from cDNA or genomic DNA of wild type (*cv*. Nipponbare), according to standard protocols and using primers indicated in Supplementary Table 3. Plasmids were constructed using Golden Gate cloning[60] as indicated in Supplementary Table 4.

**Transactivation assay**. Transactivation assays in *N. benthamiana* leaves were performed as described in the following:[61] Liquid cultures of *A. tumefaciens* Agl1, carrying the desired plasmids were set up (1 ml culture per ml used for infiltration of *N. benthaminana* leaves) and incubated at 28 °C with 180 rpm horizontal shaking. After 1.5 days the cultures were transferred to 15 ml or 50 ml Falcon tubes and spun down 30 min at 4500 rpm. The pellets were resuspended in 1 ml of *Agrobacterium* infiltration mix (AIM) (10 mM MES pH 5,6, 10 mM MgCl₂, 150 μM acetosyringone). The OD₆₀₀ value of 1:100 dilutions of the bacterial suspensions were measured and set to 1 by dilution with AIM. Then, 6 ml of a mixture were prepared by adding 1 ml of each culture carrying the desired plasmid for the respective mixture. Contained in the 6 ml per combination was also 1 ml of culture carrying the P19 plasmid, coding for a silencing suppressor. If less than five

plasmids were included in one combination, a bacterial culture carrying the placeholder plasmid pMP200 was used, to ensure that the ratio among infiltrated cultures remained the same for all treatments.

The final mixtures were incubated for 1–2 h in the dark at room temperature. Each combination was then infiltrated in three leaves of respectively four *N. benthamiana* plants of the same age and size, by injecting the bacterial solution into the bottom side of the leaf with a syringe. The plants were watered up to twice a day if required and grown two days in a climate chamber with the following conditions: 24 °C, 60% humidity, 8 h dark, 16 h light. Then, three leaf disks of 1 cm diameter were collected from each transformed leaf and transferred to 2 ml Eppendorf tubes containing respectively one glass bead, frozen in liquid nitrogen and stored at −80 °C.

One set of the leaf disks frozen in liquid nitrogen was ground in a tissue lyser with racks precooled to −80 °C, 2×1 min at 30 Hz. Then, the tubes containing the powdered plant material were placed back into liquid nitrogen. Subsequently, 300 μl of extraction buffer were added to the tubes that were then thawn on ice until the powder had dissolved in the buffer. Then, the tubes were centrifuged at 4 °C for 15 min at 15000 rpm and 170 μl of the supernatant of each tube was transferred to one well of a non-skirted low profile 96-well plate, stored on ice. To a second 96-well plate 100 μl of assay buffer were added in each well corresponding to the wells on the first plate containing the leaf protein extract. The plate with the assay buffer was then sealed and put at 37 °C. In the meantime, five black 96-well plates, labeled corresponding to 5 timepoints (*t0*, *t1 t2 t3 t4*), were filled with 100 μl of 0.2 M $Na_2CO_3$. To start the assay 10 μl of the protein extract were transferred to the corresponding wells on the plate with the assay buffer and at five predefined time points (*t0* = 0 min, *t1* = 7 min, *t2* = 14 min, *t3* = 21 min, *t4* = 28 min) 10 μl from the assay plate were transferred to the corresponding wells on the corresponding black plate. As a standard curve for the formed product 10 μl a dilution series of 4-MU (1, 0.5, 0.25, 0.125, 0.0625, 0.03125, 0.015625, and 0.0078125 μM) in extraction buffer were transferred to 100 μl of 0.2 M $Na_2CO_3$ in duplicate. To normalize the enzymatic activity to the amount of protein, a transparent 96-well plate filled with 100 ml Bradford reagent per well and 1 μl from the respective wells containing the protein extract were added. To create a standard curve for the Bradford reaction, 2 μl of a dilution series of BSA in ddH2O (5, 2.5, 1.25, 0.625, 0.3125, 0.15625, 0.078125, and 0.0390625 mg/ml) were added to 100 μl of Bradford reagent in duplicate. The composition of buffers used for the transactivation assay is presented in Supplementary Table 5.

**Root exudate collection**. Dehusked rice seeds were surface sterilized by incubating in absolute Ethanol for 2 min, then immersing twice in 3.5% Sodium hypochlorite solution for 15 min and rinsing three times with demineralized sterile water (15 min per rinse). Seeds were germinated on 1.5% 1/2 MS-agar at 22 °C in darkness for 1 week. Subsequently, germinated seedlings were exposed to light and grown for two weeks in a long-day photoperiod (16 h light/8 h dark) with 60% relative humidity in the same temperature conditions. Two seedlings were transferred to 7×7×8 cm pots filled with autoclaved sand (seven replicates of two seedlings per genotype). The plants were grown for 3 weeks in a chamber set at 22 °C in a long-day photoperiod (16 h light/8 h dark) with 60% relative humidity. Each pot was supplied with demineralized sterile water for two weeks and then with 1/2 Hoagland's solution containing 25 μM phosphate for one week. Root exudates were collected by adding and draining 150 mL water (containing 5% ethanol) through each pot. The exudates were frozen using liquid nitrogen and stored at −80 °C until use.

**Strigolactone quantification**. Isolation of strigolactones from root exudates was carried out as reported previously[24]. The analyte 4-deoxyorobanchol and the internal standard *rac*-GR24 were obtained from Olchemim (Olomouc, Czech Republic). The extracts were analyzed on a QTrap 6500+ mass spectrometer (Sciex, Darmstadt, Germany) operating in the positive electrospray ionization (ESI +) mode. The following ESI source parameters were used: ion spray voltage (5500 V), source temperature (550 °C), turbo gas (55 psi), heater gas (65 psi). MS/MS analysis was performed in the multiple reaction monitoring (MRM) mode and individual MS parameters were optimized by collision-induced dissociation after individually infusing the pure references (*rac*-GR24, 4-deoxyorobanchol) or the exudate extract (putative methoxy-5-deoxy strigol isomers) via a syringe pump into the mass spectrometer. For compound quantification, the specific mass transition of 4-deoxyorobanchol (331.0 → 185.0), putative methoxy-5-deoxy strigol isomers (361.0 → 97.0), and internal standard *rac*-GR24 (299.0 → 96.9) were used. The exudate extracts were separated on an ExionLC UHPLC system (Shimadzu, Duisburg, Germany), equipped with two ExionLC AD pump systems, an ExionLC degasser, an ExionLC AD autosampler, an ExionLC AD column oven and an ExionLC controller. Chromatographic separation was performed at a flow rate of 0.45 mL/min on 100 × 2.1 mm, 100 Å, 1.7 μm Kinetex C18 column (Phenomenex, Aschaffenburg, Germany) at 45 °C. The solvents used for chromatography were aqueous formic acid (0.1%) as eluent A and acetonitrile (0.1% formic acid) as eluent B. The following gradient was used based on reports from the literature[24] with slight modifications: 0.0–1.0 min (15% B), 1.0–5.0 min (40% B), 5.0–8.0 min (65% B), 8.0–9.0 min (65% B), 9.0–10.0 min (100% B), 10.0–11.0 min (100% B), 11.0–12.0 min (15% B), 12.0–13.0 min (15% B). The Analyst software (version 1.6.3) was used for data acquisition and processing.

**Phylogenetic analysis of PHR proteins**. *Arabidopsis* AtPHR1 was used to find close homologs in other plant species. Following this, Clustal Omega 1.2.4 (https://www.ebi.ac.uk/Tools/msa/clustalo/) was used for multiple sequence alignment (MSA) of amino acid sequences for PHR proteins and homologous MYB TFs from AM-incompetent species (*Arabidopsis thaliana* and *Brassica rapa*), monocotyledon AM-host species (*Oryza sativa*, *Zea mays*, and *Sorghum bicolor*) and dicotyledon AM-host species (*Lotus japonicus*, *Medicago truncatula*, and *Solanum lycopersicum*).

Based on the MSA, a tree was constructed using "Minimum evolution with 1000 bootstrap" in MEGA (https://www.megasoftware.net/). This tree was further modified in MEGA using "Maximum likelihood".

**Greenhouse experiment**. The soil was collected from a rice field at Longhua base in Shenzhen, China, which was inundated for a long period and then land-prepared prior to rice seeding. A long-inundated and land-prepared field was chosen to ensure soil with very low amounts of infective AM fungal propagules[41,62,63]. The soil properties were as follows: pH 5.5, total nitrogen 1.73 g·kg$^{-1}$, total phosphorus 0.72 g·kg$^{-1}$, total potassium 29.3 g·kg$^{-1}$, organic carbon 22.5 g·kg$^{-1}$, alkali hydrolysable nitrogen 216 mg·kg$^{-1}$, Olsen phosphorus 36.7 mg·kg$^{-1}$, available potassium 115.0 mg·kg$^{-1}$. The experiment was conducted in the greenhouse of Shenzhen Research Institute, Chinese University of Hong Kong, Shenzhen, Guangdong, China (22° 32 ′N, 113° 56 ′E). The temperature, sunlight, and humidity of the greenhouse used in the experiment are shown in Fig. Supplementary Fig. 16. Rice seeds were germinated and grown for 2 weeks on MS-agar. The seedlings were then transplanted to pots (each plant was put into a 4 cm × 4 cm area and 15 cm height pot) filled with collected field soil and were mock-inoculated or inoculated with *R. irregularis* (1500 spores; Agronutrition, Toulouse, France). Pure AMF spores (applied to a thin quartz sand layer) were placed at the depth of 2.5 cm from the soil surface in inoculated samples while for non-inoculated samples only a thin sand layer was included. Plants were grown in a greenhouse at around 22–32 °C (Supplementary Fig. 16) till 110 days post transplanting (dpt) and irrigated with tap water once surface soil dried so that the water layer of 2–3 cm got restored. All plants were fertilized once with urea (100 mg per pot) and potash (75 mg per pot) at 40 dpt before panicle initiation. HP samples were fertilized with superphosphate fertilizer, $P_2O_5$ at 75 mg/pot (every 20 days) while LP plants were not fertilized with superphosphate at all. Plants reached the panicle initiation stage at around 40–45 dpt and flowering started at around 75–80 dpt. At 110 days post transplanting, plant height and panicle length were determined and roots were harvested for quantification of root fresh weight, AM colonization, and gene expression analysis by RT-qPCR. Plants were air-dried for 1 week and quantified for shoot dry weight, seed setting (percentage of filled grains per total grains) and 1000 grain weight ([total grain weight per plant/total no. of grains per plant] * 1000). Further, shoots were oven-dried at 80 °C and ground to powder for shoot phosphorus determination according to the previous description[64].

**Statistics and graphics**. Data were statistically analyzed and presented graphically using R statistical environment (https://cran.r-project.org/) or Prism 8.0 (GraphPad, USA), respectively.

**Reporting summary**. Further information on research design is available in the Nature Research Reporting Summary linked to this article.

## Data availability
The RNAseq data generated in this study have been deposited in the SRA database under BioProject accession code PRJNA735781. The ChIP-Seq data generated in this study have been deposited in the SRA database under BioProject accession code PRJNA735744. The raw data for strigolactone quantification, can be accessed at https://www.ebi.ac.uk/metabolights/, reference number MTBLS4014. The rice reference transcriptome was downloaded from Phytozome v12 (https://phytozome.jgi.doe.gov/pz/portal.html). Source data are provided with this paper.

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

## Acknowledgements
We thank Prof. Mao Chuanzao (Zhejiang University; Zhejiang, China) and Prof. Chengcai Chu (Chinese Academy of Sciences; Beijing, China) for the kind donation of seeds and Philipp Chapman (Gutjahr laboratory) for excellent technical support. The study was mainly funded by research grant 2020M672839 "Phosphate regulation of rice arbuscular mycorrhiza symbiosis" by the China Postdoctoral Science Foundation and a postdoc research grant from CUHK Shenzhen Research Institute to D.D.; by the European Research Council (ERC) under the European Union's Horizon 2020 research and innovation program (grant No. 759731) 'to C.G.; by the CRC924 'Molecular mechanisms regulating yield and yield stability and plants' of the German Research Council (DFG) grant to C.G. (project B03) and C.D. (project B12); by the Natural Science Foundation of Jiangsu Province SBK2020042924 grant to M.C.; by the Hong Kong Research Grant Council grant 14177617 to J.Z. and by the Hong Kong Research Grant Council Area of Excellence Scheme grant AoE/M-403/16 grant to H.M.L. and J.Z.

## Author contributions
Conceptualization: D.D. and C.G. Methodology: D.D., M.P., M.G., and C.G. Investigation: D.D., M.P., K.H., and M.G. Visualization: D.D. with input from C.G. Funding acquisition: D.D., C.G., J.Z., H.M.L., and C.D. Project administration: D.D., M.C., C.G., J.Z., and H.M.L. Supervision: C.G., M.C., and J.Z. Writing—original draft: D.D. and C.G. Writing—methods: D.D., M.P., K.H., M.G., and C.G. Writing—review & editing: D.D. and C.G.

## Funding

## Competing interests
The authors declare no competing interests.
