## [Peer Review File · Nature Communications]

PHOSPHATE STARVATION RESPONSE transcription factors enable arbuscular mycorrhiza symbiosisREVIEWER COMMENTS

Reviewer #1 (Remarks to the Author):

Considerable number of studies attested the relevance of PHR1 in controlling the plant adaptation to phosphate scarcity. However, many of them were performed under an abiotic context, and, because that, its role in the plant-microbe interactions was overlooked for more than a decade. Recent studies have demonstrated that PHR1 is determinant component in the plant-microbe interactions, including in the symbiosis between legumes and nitrogen-fixing bacteria. Now, Das et al., provides new evidence indicating that this master regulator (PHR1 or PHR2 in rice) is also crucial for the symbiosis with AM fungi, the most ancient and important symbiosis in extant land plants. The novelty of this study is that authors provide evidence indicating that PHR2 participate in each stage of the symbiosis between rice and AM fungi. For instance, the expression of genes involved in the biosynthesis of strigolactones, perception of karrikins, development of the arbuscule, Pi transport is downregulated in *phr2* mutant plants. This observation was reinforced by the fact that PHR2 physically interacts with the promoter regions of these genes and promotes their expression. The hypothesis that PHR2 is required for the establishment of AM symbiosis and to ensure the nutritional benefits from this symbiosis was confirmed by overexpressing PHR2. The overexpression of PHR2 not only allowed rice plant to establish this symbiosis under both optimal low phosphate conditions, but also, enhanced the Pi uptake and plant biomass. The data presented in this manuscript will significantly contribute into the understanding of how plant Pi status regulates the symbiosis with AM fungi, and also provide provocative evidence that can be used to explain how plants conquer the Pi-scarce environments and the relevance of PHR1/2 in this process.

Although the experiments are well designed and most of the shown data is solid, the authors must address fundamental questions. In my opinion, answering these questions will help readers to understand the hypothesis and the reinforce of the shown data.

Comments:

1) Selected PHR2 gene

The authors mention that rice genome contains four PHR genes: PHR1-4. Based on published promoter:GUS activity data the authors focused on PHR2. Based on this already published data, PHR2 is expressed in roots and in tissues relevant for AM symbiosis. I understand the logic of this decision, no argue about that. However, this published data was obtained from rice plants growing in the absence of AM fungi. Thus, how PHR2 is expressed in response to AM? Does PHR2 has a symbiotic expression pattern? Since the authors show evidence indicating that PHR2 control the expression of genes participating in each stage of the AM symbiosis, does PHR2 has time- and tissue-dependent expression?

It is very important to experimentally answer these questions, and for that, authors must assess the promoter PHR2::GUS activity in response to AM fungi and across the different stages of this symbiosis. Having these results will provide enough information to understand how PHR2 transcriptional behavior behaves in the presence of AM fungi.

PHR is present in almost all extant plants, including non-host AM plants like *A. thaliana* and *Marchantia polymorpha*. Was PHR1/2 coopted for AM? Or was PHR part of the symbiotic tool-kit to interact with AM fungi? The authors must discuss these two questions, but also it is necessary that the authors experimentally evaluate whether *A. thaliana* PHR1 (AtPHR1) promoter has a similar or different response to AM fungi. Also, it will be interesting to see whether AtPHR1, driven by either its own promoter or rice promoter, can complement the *phr2* mutant. These last suggested experiments can provide more clues about the evolutionary history of PHR1 and will increase the novelty and significance of this study.

Although this study brings novel data supported by very well-designed experiments, they must be reinforced by additional experiments in at least one additional AM-host plant, like *Medicago truncatula* or *Lotus japonicus*. Demonstrating that the role of PHR1/2 in the AM symbiosis is not limited to rice, will increase the novelty and importance of this study. This can be done by silencing (i.e., RNAi) the expression of PHR1/2 in one of these model legumes and evaluate whether the downregulation of PHR1/2 also decreases root colonization by AM fungi.

SPX proteins are negative regulators of PHR under optimal Pi conditions. These proteins bind to PHR1 and prevent from being translocated to the nucleus. This can be a clear explanation about why the overexpression of PHR2 is translated into a drastic phenotype. The authors barely mention this in their conclusion. Thus, the authors must mention and discuss this in detail. This information is crucial to understand the role of PHR on this symbiosis.

Reviewer #2 (Remarks to the Author):

Das et al describe a carefully conducted and complete study about the mechanism that mediates inhibition of arbuscular mycorrhizal symbiosis when the host plant is well supplied with phosphate. The phenomenon has been known for nearly a century, yet the mechanisms behind them remained elusive.

Since AM involves a major transcriptional reprogramming, a central role has been predicted for TFs. In addition, recent evidence implicates an involvement of hormonal regulation by GA.

In the present study, Das et al have tested the hypothesis, that elements in the phosphate starvation response, in particular Myb TFs of the PHR type may be involved. Indeed, mutants in PHR2 are AM-defective and PHR2-overexpressors show signs of an induced transcriptional AM response. Hence, it appears that phosphate may act through repression of PHR2 activity through previously characterized interactions with SPX proteins. RNAseq and ChIPseq experiments provide a good overview over the patterns of transcriptional regulation by PHR2. This is a careful and well-documented study, the manuscript is well-written and the data convincing. There remain just some points that should be discussed:

General comments

A general point is the question how much of the DEGs in the comparison between *phr2*-AM and *wt*-AM is responding directly to the absence of *phr2*, or, indirectly, to the much lower colonization levels in *phr2*. This is particularly the case for highly AM-inducible genes, and should be discussed. The ChIPseq approach is independent of transcript levels, and can therefore give a more AM-independent picture of PHR2-dependent gene expression, although also there, the interaction with the promoters may be affected (indirectly) by AM-inducible regulators (e.g. SPX proteins), that may affect promoter-binding (although for many AM-related genes, PHR2 seems to be sufficient for induction in the absence of AM and AM-inducible genes).

The same argument holds for the issue with *smax1* mutants: Since they exhibit higher colonization levels, increased expression of DEGs that are down in *phr2* could be due indirectly to the higher degree of colonization, not (or not only) to overlapping regulation of PHR2 and karrikin signaling.

Could there be an involvement of hormones at some level of the PHR2-mediated regulation of AM? Carbonnel & Gutjahr (2014) have speculated that hormonal mechanisms could act in the same or parallel pathways to regulate AM in response to phosphate. For SL this is clearly the case since CCD7 and CCD8 are affected. Another possibility is that gibberellin (GA) is involved. Devaiah et al. (2014) reported on a connection between phosphate starvation and GA signaling, and recent evidence showed that GA may be involved in the phosphate repression of AM (Nouri et al., 2021). It would therefore be of great interest to test, whether expression of phosphate starvation genes (incl. PHR2) is affected by hormones (e.g. GA, ethylene, ABA, and auxin), or whether *phr2* mutants or PHR2-OX have altered hormone responses. This could be simply assessed by search for GA-related genes in RNAseq and ChIPseq data, or search for relevant promoter-elements in GA-related genes. Interestingly, the authors report that the GA-related SLR1/DELLA gene was identified by ChIPseq in one replicate. Since DELLA is required for AM (Floss et al., 2013), repression of PHR2 activity at high P levels could potentially impinge on AM through reduced DELLA function.

Surprisingly many genes are affected in mock-treated *phr2* mutants (Figure 2a). This means that *phr2* may have considerable AM-independent phenotypes. How do these plants look in general? Since they have a general growth phenotype (Fig S17), it would be interesting to assess them more specifically for developmental phenotypes, for example similarity with hormonal dwarf mutants. The authors suggest that PHR2 mis-expression may result in growth defects because of P-toxicity, but hormonal effects would also be possible.

Minor points:

Page 3, Line 10:

State the number of genes in the list (205)

Page 3, line 38: As it is stated now, it seems that all promoter elements are sitting exactly at -3000 from the ATG. Please rephrase to state that they are between ATG and -3000.

Page 3, Line 45-46.

Given that <1.5% of all genes are PHR2 targets (n=435 among >40'000 genes), the fact that 13% of the AM gene list are PHR2 targets (27 in 205) seems a quite significant number (ca. 10-fold enrichment). It would be relevant to mention this, and confirm it by a simple statistical test.

Page 4, line 20: Sentence seems to be incomplete.

Page 4, Line 26: REQUIRED

Page 4, Line 44:

P toxicity does not seem plausible, given the minimal increase of P content in PHR2-OX (at HP and AM) relative to wt (approximately 2.6 vs. 3.2 P content), although the actual concentrations may be slightly different even though the dry weights are very similar (Fig. S17). Hence, other consequences of transcriptional deregulation in PHR2OX may be responsible for the quite drastic growth defects (Fig. S17), e.g. a badly developed root system due to disturbed developmental programs, or deregulation of nutrient homeostasis.

Fig. 1, legend, title:

Effect of PHR2 mutation AND OVEREXPRESSION on root colonization by AM fungi

FigS17;

Since not only AM/mock should be considered, statistical analysis between genotypes should be included (indicated with letters over all treatments).

We thank the reviewers for their thoughtful comments on our manuscript.

Reviewer #1 (Remarks to the Author)

Considerable number of studies attested the relevance of PHR1 in controlling the plant adaptation to phosphate scarcity. However, many of them were performed under an abiotic context, and, because that, its role in the plant-microbe interactions was overlooked for more than a decade. Recent studies have demonstrated that PHR1 is determinant component in the plant-microbe interactions, including in the symbiosis between legumes and nitrogen-fixing bacteria. Now, Das et al., provides new evidence indicating that this master regulator (PHR1 or PHR2 in rice) is also crucial for the symbiosis with AM fungi, the most ancient and important symbiosis in extant land plants. The novelty of this study is that authors provide evidence indicating that PHR2 participates in each stage of the symbiosis between rice and AM fungi. For instance, the expression of genes involved in the biosynthesis of strigolactones, perception of karrikins, development of the arbuscule, Pi transport is downregulated in *phr2* mutant plants. This observation was reinforced by the fact that PHR2 physically interacts with the promoter regions of these genes and promotes their expression. The hypothesis that PHR2 is required for the establishment of AM symbiosis and to ensure the nutritional benefits from this symbiosis was confirmed by overexpressing PHR2. The overexpression of PHR2 not only allowed rice plant to establish this symbiosis under both optimal low phosphate conditions, but also, enhanced the Pi uptake and plant biomass. The data presented in this manuscript will significantly contribute into the understanding of how plant Pi status regulates the symbiosis with AM fungi, and also provide provocative evidence that can be used to explain how plants conquer the Pi-scarce environments and the relevance of PHR1/2 in this process.

Although the experiments are well designed and most of the shown data is solid, the authors must address fundamental questions. In my opinion, answering these questions will help readers to understand the hypothesis and the reinforce of the shown data.

Reply: We thank the reviewer for the overall positive reception of our manuscript.

Comments:

1) Selected PHR2 gene

The authors mention that rice genome contains four PHR genes: PHR1-4. Based on published promoter:GUS activity data the authors focused on PHR2. Based on this already published data, PHR2 is expressed in roots and in tissues relevant for AM symbiosis. I understand the logic of this decision, no argue about that. However, this published data was obtained from rice plants growing in the absence of AM fungi. Thus, how PHR2 is expressed in response to AM? Does PHR2 has a symbiotic expression pattern? Since the authors show evidence indicating that PHR2 control the expression of genes participating in each stage of the AM symbiosis, does PHR2 has time- and tissue-dependent expression? It is very important to experimentally answer these questions, and for that, authors must assess the promoter:PHR2::GUS activity in response to AM fungi and across the different stages of this symbiosis. Having these results will provide enough information to understand how PHR2 transcriptional behaves in the presence of AM fungi.

Reply: Thank you for this comment. Shi et al 2021 have now published the promoter activity pattern of PHR2 in rice roots colonized by AM fungi. GUS activity was observed uniformly distributed in the root cortex with a stronger signal in arbuscule-containing cells.

PHR is present almost all extant plants, including non-host AM plants like *A. thaliana* and *Marchantia polymorpha*. Does PHR1/2 was coopted for AM? Or Does PHR was part of the symbiotic tool-kit to interact with AM fungi? The authors must discuss these two questions,

but also it is necessary that the authors experimentally evaluate whether *A. thaliana* PHR1 (AtPHR1) promoter has a similar or different response to AM fungi. Also, it will be interesting to see whether AtPHR1, driven by either its own promoter or rice promoter, can complement the *phr2* mutant. These last suggested experiments can provide more clues about the evolutionary history of PHR1 and will increase the novelty and significance of this study.

Reply: We thank the reviewer for starting this interesting discussion. Our manuscript deals with the function of PHR2. We show for the first time that PHR2 regulates genes required for early stages of arbuscular mycorrhiza development such as common symbiosis genes. Our manuscript does not address the evolution of PHR's role in AM and neither the evolution of its promoter. This is beyond the scope of our manuscript and will be dealt with in another project. However, to answer another request by this reviewer we demonstrate that PHR also regulates AM symbiosis with *Lotus japonicus*, indicating that its function in AM is at least broadly conserved across Angiosperms. We would also like to mention that AM has been lost at the basis of the Brassicaceae long before Arabidopsis evolved. Studying the Arabidopsis PHR protein and *PHR* promoter in the context of AM therefore, will not tell us about co-option of PHR for AM. It will only answer the question whether promoter elements and the protein sequence have been maintained after AM was lost. Whether it has played a role in AM of early land plants can only be reconciled with mutants of for example the AM-host liverwort *Marchantia paleacea*. We do not clearly understand what the reviewer means with 'symbiotic toolkit' in the context of PHRs. PHR is already present in the unicellular alga *Chlamydomonas reinhardtii* (Rubio et al 2001) and – although we cannot exclude a secondary acquisition - has therefore likely evolved prior to AM. To accommodate the reviewer's request we now mention *Chlamydomonas* in our manuscript and also added the following text to the discussion:

"We show that the role of PHRs in AM is conserved in the model legume *L. japonicus*. It is therefore likely, that AM symbiosis is a part of the PHR-mediated phosphate starvation response syndrome at least across the angiosperms. It will be interesting to learn whether this hypothesis can be confirmed in other clades of the vascular and early-diverging non-vascular plants".

Although this study brings novel data supported by very well-designed experiments, they must be reinforced by additional experiments in at least one additional AM-host plant, like *Medicago truncatula* or *Lotus japonicus*. Demonstrating that the role of PHR1/2 in the AM symbiosis is not limited to rice, will increase the novelty and importance of this study. This can be done by silencing (i.e., RNAi) the expression of PHR1/2 in one of these model legumes and evaluate whether the downregulation of PHR1/2 also decrease root colonization by AM fungi.

Reply: Thank you for this comment. We added new data (Fig. 4 and Fig. S19), showing that mutation of *Lotus japonicus PHR1A* leads to a reduction of root colonization by *Rhizophagus irregularis* at low phosphate conditions, as well as a reduction in the expression of the common symbiosis genes *SYMRK*, *CCaMK* and *CYCLOPS*. Furthermore, ectopic expression of *PHR1A* driven by the Ubiquitin promoter leads to increase in root colonization at high phosphate. This shows that the role of PHRs in AM symbiosis is conserved in *Lotus japonicus*.

SPX proteins are negative regulators of PHR under optimal Pi conditions. These proteins bind to PHR1 and prevent from being translocated to the nucleus. This can be a clear explanation about why the overexpression of PHR2 is translated into a drastic phenotype. The authors barely mention this in their conclusion. Thus, the authors must mention and discuss this in detail. This information is crucial to understand the role of PHR on this symbiosis.

Reply: We thank the reviewer for this helpful comment and have now added the role of SPX in the discussion of our manuscript.

Reviewer #2 (Remarks to the Author)

Das et al describe a carefully conducted and complete study about the mechanism that mediates inhibition of arbuscular mycorrhizal symbiosis when the host plant is well supplied with phosphate. The phenomenon has been known for nearly a century, yet the mechanisms behind them remained elusive. Since AM involves a major transcriptional reprogramming, a central role has been predicted for TFs. In addition, recent evidence implicates an involvement of hormonal regulation by GA.

In the present study, Das et al have tested the hypothesis, that elements in the phosphate starvation response, in particular Myb TFs of the PHR type may be involved. Indeed, mutants in PHR2 are AM-defective and PHR2-overexpressors show signs of an induced transcriptional AM response. Hence, it appears that phosphate may act through repression of PHR2 activity through previously characterized interactions with SPX proteins. RNAseq and ChIPseq experiments provide a good overview over the patterns of transcriptional regulation by PHR2. This is a careful and well-documented study; the manuscript is well-written and the data convincing. There remain just some points that should be discussed:

Reply: We thank the reviewer for the positive reception of our findings.

General comments

A general point is the question how much of the DEGs in the comparison between *phr2*-AM and *wt*-AM is responding directly to the absence of *phr2*, or, indirectly, to the much lower colonization levels in *phr2*. This is particularly the case for highly AM-inducible genes, and should be discussed. The ChIPseq approach is independent of transcript levels, and can therefore give a more AM-independent picture of PHR2-dependent gene expression, although also there, the interaction with the promoters may be affected (indirectly) by AM-inducible regulators (e.g. SPX proteins), that may affect promoter-binding (although for many AM-related genes, PHR2 seems to be sufficient for induction in the absence of AM and AM-inducible genes).

The same argument holds for the issue with *smax1* mutants: Since they exhibit higher colonization levels, increased expression of DEGs that are down in *phr2* could be due indirectly to the higher degree of colonization, not (or not only) to overlapping regulation of PHR2 and karrikin signaling.

Reply: We agree with the reviewer that the expression level of AM-induced genes strongly depends on the level of root colonization. This is the reason why we mainly focused on the difference in transcript accumulation in non-colonized roots (Figure 2C-F). Also, the DEGs from the *smax1* mutant are from non-colonized roots (Choi et al 2020). We also agree that for the ChIP-seq we probably mainly detected promoters, for which no AM-induced transcription factor is needed to stabilize the interaction of PHR2 with the DNA. We have now added a new figure (Fig. S7) to show that there is a considerable overlap between genes that are down in *phr2* vs. WT in non-colonized roots with genes that are also less expressed in *phr2* vs. WT in colonized roots.

Could there be an involvement of hormones at some level of the PHR2-mediated regulation of AM? Carbonnel & Gutjahr (2014) have speculated that hormonal mechanisms could act in the same or parallel pathways to regulate AM in response to phosphate. For SL this is clearly the case since CCD7 and CCD8 are affected. Another possibility is that gibberellin (GA) is involved. Devaiah et al. (2014) reported on a connection between phosphate starvation and GA signaling, and recent evidence showed that GA may be involved in the phosphate repression of AM (Nouri et al., 2021). It would therefore be of great interest to test, whether expression of phosphate starvation genes (incl. PHR2) is affected by hormones (e.g., GA, ethylene, ABA, and auxin), or whether *phr2* mutants or PHR2-OX have altered hormone responses. This could be simply assessed by search for GA-related genes in RNAseq and ChIPseq data, or search for relevant promoter-elements in GA-related genes. Interestingly, the authors report that the GA-related SLR1/DELLA gene was identified

by ChIPseq in one replicate. Since DELLA is required for AM (Floss et al., 2013), repression of PHR2 activity at high P levels could potentially impinge on AM through reduced DELLA function.

Reply: Thank you for this comment. We have now measured strigolactones in the exudates of *phr2* and found that less strigolactones are exuded as compared to WT (Fig. 3F). We have also looked for GA-related genes in the DEGs that are less expressed in *phr2* vs. WT in non-colonized roots. We find a significant enrichment of GA-related genes in this gene set, indicating that GA-metabolism and signalling may be misregulated in *phr2*. We also discuss the involvement of GA in suppression of AM under high phosphate and cite Nouri et al 2021.

Surprisingly many genes are affected in mock-treated *phr2* mutants (Figure 2a). This means that *phr2* may have considerable AM-independent phenotypes. How do these plants look in general? Since they have a general growth phenotype (Fig S17), it would be interesting to assess them more specifically for developmental phenotypes, for example similarity with hormonal dwarf mutants. The authors suggest that PHR2 mis-expression may result in growth defects because of P-toxicity, but hormonal effects would also be possible.

Reply: Thank you for this comment. The AM-independent phenotypes of *phr2* and 35S:PHR2 have been previously described.

- Lv, Qundan, et al. "SPX4 negatively regulates phosphate signaling and homeostasis through its interaction with PHR2 in rice." *The Plant Cell* 26.4 (2014): 1586-1597.
- Zhou, Jie, et al. "OsPHR2 is involved in phosphate-starvation signaling and excessive phosphate accumulation in shoots of plants." *Plant Physiology* 146.4 (2008): 1673-1686.
- Guo, Meina, et al. "Integrative comparison of the role of the PHOSPHATE RESPONSE1 subfamily in phosphate signaling and homeostasis in rice." *Plant Physiology* 168.4 (2015): 1762-1776.

We agree that hormonal effects can also be possible reasons for the dwarf phenotype of 35S:PHR2. We now mention this possibility in the manuscript text.

Minor points:

Page 3, Line 10:

State the number of genes in the list (205)

Reply: Done.

Page 3, line 38: As it is stated now, it seems that all promoter elements are sitting exactly at -3000 from the ATG. Please rephrase to state that they are between ATG and -3000.

Reply: Done.

Page 3, Line 45-46.

Given that <1.5% of all genes are PHR2 targets (n=435 among >40'000 genes), the fact that 13% of the AM gene list are PHR2 targets (27 in 205) seems a quite significant number (ca. 10-fold enrichment). It would be relevant to mention this, and confirm it by a simple statistical test.

Reply: We have done this and added the result to a new figure (Fig. S15). Probably the reviewer meant the 27 (8 + 4 + 15) genes in Fig. 3A which are PHR2 targets in single and both ChIP-Seq reps and also overlap with genes downregulated in *phr2* vs WT and AM genelist. However, to use high-confidence targets we have considered the 17 (15 +2) genes in Fig. 3A common to both ChIP-Seq reps and the AM genelist for the enrichment analysis.

Page 4, line 20: Sentence seems to be incomplete.

Reply: Thank you the sentence was corrected.

Page 4, Line 26: REQUIRED

Reply: Corrected.

Page 4, Line 44:

P toxicity does not seem plausible, given the minimal increase of P content in PHR2-OX (at HP and AM) relative to wt (approximately 2.6 vs. 3.2 P content), although the actual concentrations may be slightly different even though the dry weights are very similar (Fig. S17). Hence, other consequences of transcriptional deregulation in PHR2OX may be responsible for the quite drastic growth defects (Fig. S17), e.g. a badly developed root system due to disturbed developmental programs, or deregulation of nutrient homeostasis.

Reply: In the previous version of our manuscript, we have shown the total P uptake. We have now added also the P concentration (Figure S22A). The P concentration in shoots of *35S:PHR2* lines is higher than for WT and *phr2* mutants. However, the reviewer is right that misregulation of genes involved in regulating plant growth is also a possible reason. We included this possibility into the manuscript.

Fig. 1, legend, title:

Effect of PHR2 mutation AND OVEREXPRESSION on root colonization by AM fungi

Reply: Thank you, we corrected this.

FigS17;

Since not only AM/mock should be considered, statistical analysis between genotypes should be included (indicated with letters over all treatments).

Reply: We performed the statistical analysis across all comparisons.

REVIEWERS' COMMENTS

Reviewer #1 (Remarks to the Author):

I have read the revised version of this manuscript. The authors have successfully addressed all my comments. Indeed, the new version is solid and convincing. This revised version also integrates the current knowledge about the role of PHR and SPX proteins in the establishment of AM symbiosis. I don't have further concerns or comments for this manuscript, other than this paper will advance our knowledge about the regulation of AM symbiosis by PHR.

Reviewer #2 (Remarks to the Author):

The authors answered satisfactorily to the comments and criticisms and performed the requested revisions.